# Power Transformer Voltages Classification with Acoustic Signal in Various Noisy Environments

**DOI:** 10.3390/s22031248

**Published:** 2022-02-07

**Authors:** Mintai Kim, Sungju Lee

**Affiliations:** Department of Software, Sangmyung University, Cheonan 31066, Korea; 201820985@sangmyung.kr

**Keywords:** power transformer, acoustic signal, noise rejection, ensemble model

## Abstract

Checking the stable supply voltage of a power distribution transformer in operation is an important issue to prevent mechanical failure. The acoustic signal of the transformer contains sufficient information to analyze the transformer conditions. However, since transformers are often exposed to a variety of noise environments, acoustic signal-based methods should be designed to be robust against these various noises to provide high accuracy. In this study, we propose a method to classify the over-, normal-, and under-voltage levels supplied to the transformer using the acoustic signal of the transformer operating in various noise environments. The acoustic signal of the transformer was converted into a Mel Spectrogram (MS), and used to classify the voltage levels. The classification model was designed based on the U-Net encoder layers to extract and express the important features from the acoustic signal. The proposed approach was used for its robustness against both the known and unknown noise by using the noise rejection method with U-Net and the ensemble model with three datasets. In the experimental environments, the testbeds were constructed using an oil-immersed power distribution transformer with a capacity of 150 kVA. Based on the experimental results, we confirm that the proposed method can improve the classification accuracy of the voltage levels from 72 to 88 and to 94% (baseline to noise rejection and to noise rejection + ensemble), respectively, in various noisy environments.

## 1. Introduction

The transformer is one of the transmission and distribution facilities of power systems. Transformer mechanical failure accounts for 77% of the on-load tap changers (OLTCs), windings, and cores [1]. The traditional methods for diagnosing the mechanical failure of a transformer mainly measure vibration using a vibration sensor installed on the box wall of the transformer, and analyze the characteristic information, such as winding deformation or looseness defects [2,3,4,5,6,7,8,9,10,11,12,13,14]. The method using the vibration sensor belongs to the laminating type of detection, and there are specific requirements for the installation, location, and measurement methods by using vibration sensors, such as the accelerometer sensor, to improve the reliability of the vibration signal. On the other hand, since the over-voltage supplied to the transformer causes the overheating and deformation of the windings to result in failure, it is important to determine the abnormal supply voltage levels that cause the failure of the transformer. Recently, methods for the monitoring and fault diagnosis of transformers have been studied as advanced and complex methods [15,16,17,18,19,20].

The online monitoring method for the transformer should not interfere with the normal operation of the transformer, nor should it require a special operating mode for monitoring. To satisfy these requirements, some studies have recently reported on the monitoring of the mechanical condition of transformers and diagnosis of faults by measuring acoustic signals to ensure safe and reliable transformer operation [15,16,17,18,19,20]. Note that since the acoustic signal is generated by the vibration of the windings and cores of the transformer, the acoustic signal contains sufficient information to determine the transformer state. The acoustic signal is propagated through the air, and thus, the acoustic signals can be easily collected by a microphone sensor or recording equipment, which provides a convenient technique using a non-contact method, compared to the existing vibration measurement method. Thus, since the acoustic signal-based method can be applied in a non-invasive way, it is suitable as a method that does not interfere with the normal operation of the transformer. However, since the acoustic signal-based monitoring system is affected by the surrounding noise environment, we need a classification method that is robust against noise for realistic application.

In this study, we propose a method for monitoring the transformer status by classifying over-, normal-, and under-voltage levels based on the acoustic signal of the transformer operating in various noisy environments. In particular, we focus on the audible frequency band, which can be recognized by humans from the acoustic signal; that is, the acoustic signal is measured within the audible frequency, and the measured acoustic signal is converted into a Mel Spectrogram (MS) [21,22,23]. To design the classification model, we exploit the idea of the U-Net [24] encoder layers to extract and express the important features from the acoustic signal. For robustness against the surrounding noise, the noise is rejected by using deep learning-based U-Net [25]. In the case of U-Net, the predefined noise dataset is required. In this study, a noise dataset is constructed using nature (e.g., rain and wind), worksite (e.g., heavy equipment and welding), and city (e.g., vehicles and crowds) noises. 

However, it is difficult to design U-Net as a robust classification model against an unknown noise due to the dependence on the known noise dataset (i.e., the predefined dataset). In the unknown noise environment, we design an ensemble model of the bagging method [26] to avoid accuracy degradation. The ensemble model is designed by using three datasets with noise-free, the noise-containing, and noise-reducing audible acoustic signals, respectively. The contributions of the proposed method are summarized as follows:Using a non-invasive method with the audio sensor and the acoustic signal in the audible frequency band to simulate the environment of human hearing, the over-, normal-, and under-voltage levels are classified according to the voltage supplied to the operating transformer. The classification model is designed based on the U-Net encoder layers to extract and express the important features from the acoustic signal.In the known noisy environment, the noise rejection method is designed by using U-Net-based deep learning to reduce noise. A predefined noise dataset, constructed by using nature (e.g., rain and wind), worksite (e.g., motor and welding), and city (e.g., vehicles and crowds) noises, is used.The ensemble model is designed to improve the classification accuracy in the unknown noisy environment. we exploit a concept of an ensemble technique that can be robust against unknown noises by improving the generalization performance of deep learning models through model diversity.

In the experimental environments, the testbeds are constructed using an oil-immersed power distribution transformer with a capacity of 50 kVA, and the supply voltages are set at 90, 95, 105, and 110% based on 22,900 V/380 V (100%). The built-in microphone of a smartphone (Galaxy S8) is used to collect sound signals. As a result of the experiment, we confirm that the proposed method can improve the voltage level classification accuracy from 72 to 88 and to 94% (baseline to noise rejection and to noise rejection + ensemble), respectively, in a noisy environment.

This paper is organized as follows: Section 2 summarizes previous vibration–acoustic signal-based transformer mechanical fault detection methods. Section 3 describes the proposed method with the deep learning-based voltage level classification, which is robust in the known and unknown noise environments. Section 4 explains the details of the experimental results, while Section 5 concludes the paper. 

## 2. Background

### 2.1. Vibration–Acoustic Signal-Based Transformer Mechanical Fault Detection

The transformer mechanical failure accounts for 77% of OLTCs, windings, and cores [1]. The authors of [2] proposed winding condition management based on principal component analysis by using the mathematical model of transformer core/winding vibrations from [3,4,5,6,7,8,9,10]. The vibration depends on the power between the fundamental voltage and current; thus, the currents and voltages must be treated as complex variables [11,12,13,14]. The previous methods for diagnosing the mechanical failure of winding cores mainly measure vibration using an accelerometer sensor installed on the box wall of the transformer, and extract and analyze the characteristic vibration information for the transformer mechanical failures, such as winding deformation or looseness defects [2,3,4,5,6,7,8,9,10,11,12,13,14,15]. 

In general, the vibration data can be analyzed in the time, frequency, and time–frequency domains. In the time domain, the magnitude of a signal for a certain time is analyzed. In the frequency domain, the distribution of each frequency is analyzed [2,3,4,5,6,7,8,9,10,11,12,13,14,15,16,17,18,19,20]. Additionally, in the time–frequency domain, the vibration and amplitude of the frequency according to a certain time are analyzed. Geng [18] used the time–frequency spectrogram of the acoustic signal and AlexNet [27] as a method for diagnosing the mechanical failure of the dry-type transformer. Previous methods have often used the acoustic vibration signal to monitor and analyze the mechanical faults, such as winding deformation or loosening defects [15,16,17,18,19,20], which has strong anti-interference and high sensitivity advantages. However, the vibration signal is usually obtained through the vibration sensor placed on the box wall of the transformer. This technique belongs to the laminating type detection, and has certain requirements for the installation position and method of the vibration sensor to improve the reliability of the front-end vibration signal detection. The acoustic signal is obtained by the vibration of the transformer through the air during transformer operation, and the vibration and the acoustic signal are interrelated and homologous. In addition, the acoustic signal can be easily collected by a microphone sensor or recording equipment, which belongs to non-contact measurement, and has the advantages of simplicity and convenience. Recently, research reports of transformer mechanical condition monitoring and analysis with the acoustic signal instead of the vibration have been reported [15,16,17,18,19,20]. 

In recent years, the machine learning model has been studied by the deep convolution neural network. It simulates the learning mechanism of the biological neural network, extracts the hidden features from a low to high level, and finally completes the classification, such as the transformer faults. Deep learning techniques are widely used in speech recognition and image processing, and have achieved good results in mechanical fault diagnosis and power quality classification [18,19,20]. Table 1 shows the existing methods for the fault diagnosis of the windings and cores of transformers based on vibration and acoustic signals [2,3,4,5,6,7,8,9,10,11,12,13,14,15,16,17,18,19,20,21]. In this study, the voltage levels of the transformer were classified based on the acoustic signal and deep learning techniques considering the various noise environments. Note that this study assumed a power distribution transformer, while OLTC failure was not considered. Therefore, we focused on the analysis of the acoustic signal according to the voltage levels in relation to the distribution transformer failure, such as of the winding or core.

### 2.2. Noise Rejection Method with U-Net

U-Net has been introduced into biomedical imaging techniques to improve the precision and location of microscopic images of neuronal structures [24]. U-Net consists of an encoder (i.e., Conv2D) and a decoder (i.e., Deconv2D) by using convolution layers with various convolutional filters. In the U-Net encoder, the Conv2D layer, which reduces the image size by half and doubles the number of channels, is used to encode the image into a small and deep representation. Then, in the U-net decoder, the encoded result is decoded to the original image size through DeConv2D layers, which half the channel size and double the image size. At this time, in the decoding process, a skip connection that concatenates the previously encoded result is used to preserve a high level of detail (it is crucial that the reproduction preserves a high level of detail). This means that the encoded results represent the more relatively important information, while reducing the unimportant information (i.e., noise). Jansson et al. [25] exploit the idea of U-Net to tasks of audio speech separation from image processing. Figure 1 shows the U-Net architecture. 

Although previous approaches [18,19,20] have studied the diagnosis of various mechanical failures, these studies have focused on the classification method or considered the known noise environments. Therefore, we need a way to consider the unknown noise as well as the known noise. 

### 2.3. Acoustic Signal Analysis in Time–Frequency Domain

To analyze the acoustic signal, the Linear Prediction Cestrum Coefficient (LPCC) [21], Gammatone Filter Cepstral Coefficient (GFCC) [22], and Mel Frequency Cestrum Coefficient (MFCC) [23] are widely used in the time–frequency domain. LPCC can represent the acoustic signal as the frequency for a certain time without signal distortion. GFCC has the advantage of representing a higher frequency than the other methods. MFCC is used to represent human speech from the relative low frequency of the acoustic signal. The human auditory system has different perception abilities for speech signals, such as different frequencies. The human auditory system has a good voice recognition ability, robustness, and an anti-noise ability, and can effectively improve the accuracy of speaker recognition in noisy and unstable environments. Considering that the transformer acoustic signal has certain commonality with the speech signal, it is possible to explain characteristic parameters based on human auditory characteristics into the transformer acoustic signal analysis. In this study, we analyzed acoustic signals that can be discerned by human hearing. Therefore, we chose a method to express the characteristics of the transformer acoustic signals using MS that mimics the human cochlea [23]. The spectrogram is a tool for visualizing and understanding sound or waves, and combines the characteristics of a waveform and a spectrum. 

## 3. Proposed Method

In this study, we propose a method to determine the transformer state by classifying the over-, normal-, and under-voltage levels using the acoustic signal in various noisy environments. In particular, to classify the acoustic signals without the degradation of classification accuracy against an unknown noise environment, an ensemble model was designed by using three datasets, the noise-free, the noise-containing, and the noise-reducing transformer acoustic signals, respectively. Figure 2 shows an overview of the proposed method.

### 3.1. Deep Learning Model for Voltage Level Classification 

The acoustic signal of the transformer was measured within the audible frequency, and then the measured acoustic signal was converted into an MS using the CNN-based deep learning model. MS is used as a method to analyze the frequency component for a certain time by converting the input signal into the time–frequency domain [28]. To convert the acoustic signal of the transformer recorded at the sampling rate of 44.1 kHz to MS, the Mel-Spectrogram module of librosa [29] was used with sampling rate (i.e., *sr*), number of mel (i.e., *n_mels*), cut size (i.e., *n_fft*), and cut interval (i.e., *hop_lenth*) parameters. *sr* is the sampling rate and denotes the number of data sampled per second. *n_mels* and *n_fft* denote how many values are used to express the frequency resolution and the cut size when cutting the acoustic signal into pieces. *hop_lenth* is used as the cut interval when cutting the voice into pieces. In this study, *sr*, *n_mels*, *n_fft*, and *hop_lenth* were set to 44,100, 128, 1024, and 512, respectively, to convert a 6-s-long acoustic signal into an MS of 128 × 512, which is a size suitable for a deep learning model. Figure 3a is an acoustic signal of a transformer composed of amplitude and time axes, and Figure 3b shows an example of the transformer’s MS converted into the time–frequency domain. MS is composed of time (i.e., a unit of time in seconds), Hz (i.e., a frequency), and the magnitude (i.e., dB) of the sound amplitude for the periodically vibrating wave. 

After the acoustic signal of the transformer was converted to MS, we classified the voltage level by using a deep learning technique. Note that the CNN-based model exploits the idea of the U-Net encoder layers to extract and express the important information of the acoustic signal. Figure 4 shows the proposed classification model based on the U-Net encoder model. The network architecture of the model for classifying abnormal voltage levels uses the area corresponding to the encoder layers of U-Net. The classification model of this study consists of six convolution layers of the encoder part of U-Net and three fully connected networks (i.e., FCN). MS receives input as 128 × 512; passes each 16, 32, 64, 128, 256, and 512 convolution filter; and uses FCN consisting of 8192, 1024, 512, and 5 (i.e., five classification voltage levels) nodes to classify the voltage levels. Note that the U-Net encoder extracts and expresses efficiently from the acoustic signal. The compressed data of the encoder layers contains important features about the corresponding input data with the loss of relatively less important information. Therefore, the U-Net encoder can efficiently provide the result of the expression of the transformer acoustic signal. Hence, we designed the classification model of the transformer acoustic signal using the concept of the U-Net encoder.

The convolution layer and the three FCN layers use the Rectified Linear Unit (ReLU) as the activation function to avoid the gradient vanishing problem in the learning process, and the last layer uses softmax to increase the classification. We used categorical cross entropy for classification voltages and an adaptive moment estimation optimizer (i.e., adam) [30] for the optimization of learning.

ReLU is a well-known function to solve the gradient vanishing problem in learning processing. To avoid the gradient vanishing, we used the ReLU as an activation function in each convolution layer by using Equation (1). In ReLU, if *x* is greater than or equal to 0, *x* is returned without any change. In contrast, if *x* is less than 0, 0 is returned.
(1)$$f\left(x\right)\;=\;\left\{\begin{array}{c} \;0,\;\;x<0\\ x,\;\;x\geq 0 \end{array}\right.\;\;$$

In the last layer (i.e., the output layer), we used softmax as an activation function for the purpose of multi-class classification. The formula for softmax is shown in Equation (2). Note that softmax makes the sum of return values 1 for the total number of classes.
(2)$$f{\left(s\right)}_{i}=\frac{e^{s_{i}}}{\sum_{j}^{C}e^{s_{j}}}\;$$

In the loss function for the voltage level classification, categorical cross entropy was used by using Equation (3). The formula for categorical cross entropy is shown in Equation (3).
(3)$$CE=-\overset{C}{\underset{i}{\sum }}t_{i}\log\left(f{\left(s\right)}_{i}\right)\;$$

Finally, we used adam, which is an optimization algorithm, to find the minimum value of the loss function. The easy implementation of adam is well known as it requires less memory than other optimization algorithms [30].

Algorithm 1 shows a U-Net encoder-based deep learning model for voltage level classification. In Step 1, the size of the input value is specified as 128 × 512 × 1. In Step 2, the number of layers and filters were declared as *n_layers* and *n_filters* to express six layers, respectively. Note that kernel size, strides, and padding were used as hyperparameters to constitute the layer. *n_filters* is the number of filters used in the layer. In the model, we set the *n_filters* to increase by a multiple of 2 as the layer deepens, starting with 16 lowest layers. To design the CNN model, we set the hyperparameters with a 5 *×* 5 kernel size (i.e., *kernel_size*), 2 strides (i.e., *strides*), and 2 padding (i.e., *padding*). In Step 3, three fully connected network layers (i.e., FCNs) were configured by using flatten and dense functions. Additionally, we used the dropout function to avoid the overfitting problem in deep learning, and batch normalization was used to solve the internal covariate shift problem. In Step 4, finally, the optimizer and loss function were used as adam and categorical cross entropy, respectively.
**Algorithm 1.** Voltage level classificationInput: acoustic signalOutput: Voltage levelsStep 1Step 2Step 3Step 4input_shape = [128, 512, 1]*X* = model_input*n_layers* = 6FILTERS_LAYER = 16for i in range(*n_layers*): *n_layers* = FILTERS_LAYER × (2^i^) *X* = Conv2D(*n_layers*, *kernel_size* = (5,5), *strides* = (2,2), *padding* = 2)(*X*) *X* = BatchNormalization(momentum = 0.9, scale = True)(*X*) *X* = get_activation(activation)(“RelU”)*X* = Flatten()(*X*)*X* = Dense(8192)(*X*)*X* = Dropout(0.3)(*X*)*X* = Dense(1024)(*X*)*X* = Dropout(0.3)(*X*)*X* = Dense(512)(*X*)*X* = Dense(5)(*X*)*X* = get_activation(“softmax”)(*X*)model = Model(model_input, *X*)model.compile(optimizer = adam(lr = 0.0001),       loss= ‘categorical_crossentropy’,metrics = [‘accuracy’])

### 3.2. Noise Rejection Method for Voltage Level Classification

To reduce the noise, we used the U-Net based noise rejection method. The noise rejection method with U-Net was generated by using the loss function with Equation (4) to learn the input (noise-containing acoustic signal) and output (noise-free acoustic signal). Note that the output data of U-Net are defined using f(*X*, *θ*)**X*, where *X* and *θ* are represented by the input data and the learning parameter set, respectively. In this case, f(*X*, *θ*) is a noise rejection mask. Finally, the noise rejection method could generate an acoustic signal without noise.
(4)$$L\left(X,Y,\theta \right)\;=f(X,\theta )*X-Y$$

Algorithm 2 describes the noise rejection method based on U-Net. In the model, the input and the output are the noise-containing and the noise-free acoustic signals, respectively. In the encoder process (i.e., Step 1 and Step 2), the convolution layers were similar to Algorithm 1, but the learning parameters of each convolution layer in the encoder were transmitted to de-convolution layers in the decoder to prevent a high level of detail with a skip connection by using concatenation. In Step 3 of the decoder part, dropout was used only for the first 3 layers to avoid the overfitting problem. In Step 4, a noise rejected MS was generated by multiplying the finally generated noise rejection mask with the input value. Finally, the optimizer and loss function were applied by using adam and MSE (i.e., mean absolute error) in Step 5.
**Algorithm 2.** Noise rejection methodInput: Noise-containing signalOutput: Noise-reducing signalStep 1Step 2Step 3Step 4Step 5input_shape = [128, 512, 1]*X* = model_input*n_layers* = 6FILTERS_LAYER = 16for i in range(*n_layers*): *n_layers* = FILTERS_LAYER × (2^i^) *X* = Conv2D(*n_layers*, *kernel_size* = (5,5), *strides* = (2,2), *padding* = 2)(*X*) *X* = BatchNormalization(momentum = 0.9, scale = True)(*X*) *X* = get_activation(activation)(“RelU”)for i in range(*n_layers* − 1): dropout = not (i == 1 or i == 2 or i == 3) if i > 0:  *X* = Concatenate(axis = 3)([*X*, x_encoder_layers[*n_layers* −i−1]])  *n_filters* = encoder_layer.get_shape().as_list()[−1]//2 *X* = Conv2DTranspose(*n_filters*, *kernel_size* = (5,5), *padding* = ”same”, *strides* = (2,2))(*X*) *X* = BatchNormalization(momentum = 0.9, scale = True)(*X*) if dropout:  *X* = Dropout(0.5)(*X*)  *X* = get_activation(“ReLU”)(*X*)*X* = Concatenate(axis = 3)([*X*, x_encoder_layers[0]])*X* = Conv2DTranspose(*n_filters*, *kernel_size* = (5,5), *padding* = 2, *strides* = (2,2))(*X*)*X* = get_activation(“Sigmoid”)(*X*)outputs = multiply([model_input, *X*])model = Model(model_input, outputs)model.compile(optimizer = ”adam”, loss= ‘mean_absolute_error’)

Figure 5 shows the results of the acoustic signal by using the noise rejection method. Figure 5a,b show the transformer acoustic signal and the noise. Figure 5c shows a transformer acoustic signal containing noise, and finally, Figure 5d shows the noise-reducing signals. Although the noise was not rejected completely, it could be confirmed that the noise was clearly suppressed.

Figure 6 shows the classification of voltage levels in the noise environment with the noise rejection method based on U-Net, that is, after the transformer signal containing noise is converted to MS, and the noise is rejected by using U-Net. Finally, the voltage level was classified using the proposed U-Net encoder-based CNN model. Note that the voltage level classification model was trained by using the noise-containing signal.

### 3.3. Ensemble Model in an Unknown Noise

Although the voltage level classification method based on noise rejection can avoid the degradation of accuracy by reducing the known noise, we need to consider the unknown noise as well as known noise to improve the accuracy in various noise environments. Figure 7 shows an example of the comparison of noise-reducing signals in the known and unknown noise, respectively. The known noise was more reduced than the unknown noise, as shown in Figure 7a,b. Since deep learning-based noise rejection methods such as U-Net use only the noise-free and the known noise-containing signal to train and generate the deep learning model, it is difficult to avoid the degradation of accuracy in the known noise environments. To solve the problem, we exploited an ensemble technique that is robust against unknown noise by improving the generalization performance of deep learning models through model diversity.

The ensemble model of the proposed method is based on a bagging (Bootstrap Aggregation) technique by which ensembles models are trained on different training data [26]. Bagging is a method of aggregation after training each model by Bootstrapping (selecting several times) a sample in the training data. We used three classification network models (Model A, Model B, and Model C) to design the ensemble model, as shown in Figure 8. Note that the three classification network models were designed with the same structure, but trained by three different types of data. Models A, B, and C were trained by using the noise-free, noise-containing, and noise-reducing transformer acoustic signals, respectively. In this case, the ensemble model can improve the generalization performance by using different types of data, and thus, it is robust against the unknown noise.

The proposed ensemble model is shown in Algorithm 3. In Step 1, each piece of data was used to train Model A, Model B, and Model C, respectively. In Step 2, the output values of each model were defined as PredictA, PredictB, and PredictC. In Step 3, the three output values in MergeABC were concatenated. In Step 4, the fully connected network was designed to classify five voltage levels by inputting the concatenated MergeABC. Note that in Step 5 of the EnsembleABC model, adam and categorical cross entropy were used as the optimizer and the loss function, respectively. **Algorithm 3.** EnsembleABC modelInput: Model A, B, COutput: Voltage levelsStep 1Step 2Step 3Step 4Step 5ModelA: Noise-free signalModelB: Noise-containing signalModelC: Noise-reducing signalPredictA = ModelA.ouputPredictB = ModelB.ouputPredictC = ModelC.ouputMergeABC = Concat(PredictA, PredictB, PredictC)EnsembleABC = Dense(Voltages levels = 5)(MergeABC)EnsembleABC.compile(optimizer = adam, loss = ’categorical_crossentropy’       , metrics = [‘accuracy’]

## 4. Experimental Results

### 4.1. Experimental Environments

To construct the testbeds, we used the built-in microphone of a smartphone (Galaxy S8) in a total of five states of 90, 95, 105, and 110% based on 22,900 V/380 V (100%) of the oil-immersed power distribution transformer with 150 kVA capacity and 44.1 record at a kHz sample rate, as shown in Figure 9a. The built-in microphone of the smartphone is optimized for collecting human voices, and it is suitable for collecting audible frequencies (20 Hz~20 KHz). Additionally, the noise was suppressed by applying active noise canceling based on a microphone array. Although the precise acoustic signal with special and high-performance microphones can provide better performance in the processing machine fault diagnosis, this study focused on the human audible range. The transformer acoustic signal was collected by using the built-in microphone, and a housing box was designed to attach the smartphone to an actual transformer in a non-invasive way, as shown in Figure 9b. Figure 9c shows how to collect acoustic data by attaching a smartphone. The acoustic signal data were cut into 6 s lengths to obtain a 128 × 512 size MS. Finally, by setting the *sr* of 44,100, *n_fft* of 128, *hop_lenth* of 512, and *n_mels* of 128, an MS size of 128 × 512 was generated, and used as an experimental dataset.

The transformer acoustic signal was measured in a noise-free room, one sample was generated every 6 s, and the dataset was composed of a total of 60,000 data. The noise dataset used 30 types of known and unknown noises, respectively. Furthermore, 70% of the dataset was used as training data, while the remaining 30% was used as test data. Table 2 shows the details of the known and unknown noise datasets. The learning data for noise rejection consisted of synthesized data, which were the acoustic signal of the transformer and the noise signal. In this study, the acoustic signal of the transformer and the noise sound were synthesized so that the signal-to-noise ratio (SNR) became 0, as shown in Equation (6):(5)$$SNR_{dB}=20\log_{10}\frac{A_{signal}}{A_{noise}}\;$$

In this study, three different models, baseline, denoising, and ensemble methods, were constructed according to the learned data. Table 3 shows three scenarios with the classification method, learning data, and noisy environment. S1 was the straightforward scenario as it was baseline with noise-free, and used two test data for the noise-free and noise-containing environments. S2 was the denoising scenario, which used the predefined data consisting of noise-free and known noise environments. Finally, S3 was the proposed ensemble method with three datasets. S2 and S3 were tests in the known and unknown noisy environments, respectively.

To analyze the acoustic signal, the RTX 2060 Super i7-9700 k was applied. For the three models used in the experiment, 128 batch sizes were commonly used, adam was used as an optimization function with a learning rate of 0.0001, and categorical cross entropy for multiple classification was used as a loss function. The baseline model learned 9000 MS of acoustic signals in the noisy environment with the network configuration of the proposed CNN model. The denoising model was used as the network configuration of the proposed CNN model by synthesizing the noise data with SNR 0 to achieve a noise-free acoustic signal.

To calculate the denoising performance with U-Net, $Denoise_{score\;}$ was used. x was the transformer acoustic signal with noise, y was the transformer acoustic signal without noise, $x_{denoise}$ was the transformer acoustic from which noise had been removed by the U-Net, and n was the total number of samples.
(6)$$Denoise_{score\;}=\frac{\sum_{1}^{n}\left(x-y\right)\;+\;1-\sum_{1}^{n}\left(x_{denoise}-y\right)\;\;}{\sum_{1}^{n}\left(x-y\right)\;+\;1}$$

To calculate the accuracy performance, the classification performance was used as an indicator for the voltage levels classification by using Accuracy. *TP*, *TN*, *FP*, and *FN* are true positive, true negative, false positive, and false negative, respectively; that is, *TP* and *TN* are the numbers of data in which the positive and negative predictions are correct, whereas *FP* and *FN* are numbers that do not match the actual predictions.
(7)$$Accuracy=\frac{TP+TN}{TP+TN+FP+FN}\;$$

### 4.2. Performance of Noise Rejection with U-Net

To verify the U-Net, the performance with noise rejection was validated by dividing it into known and unknown noise. In addition, the noise rejection performance was confirmed by increasing the number of predefined data using known noise. Additionally, the predefined data were classified by dividing the noise environment into nature, worksite, and city environments. Table 4 shows the noise rejection performance in each noise environment and with predefined data according to the number of known noises. Overall, it was confirmed that the higher the number of predefined noise types, the better the noise rejection performance. In the comparison of known noise and unknown noise, it was confirmed that the noise rejection performance of U-Net in an unknown noise environment was degraded, compared to that in a known noise environment. In the nature, workplace, and city noise environments assumed in this study, the noise rejection performance was relatively high in the city environment.

In addition, to confirm the classification accuracy of the transformers of S1 (baseline) and S2 (noise rejection) in the detailed noise environments, the accuracy was measured in each environment (i.e., single noise). Figure 10 shows the classification accuracy of the transformer voltage levels of both known and unknown noises after each denoising with 30 known noise types. Note that we selected 18 noises for each noise environment to compare the performance of S1 and S2. In the detailed noises, we considered the known noise (i.e., Cicadidae, rain, wind, frill, factory alarm, construction site, car horn, main street, and car), and the unknown noise (i.e., bird, typhoon, hail, welding, grinder, clothing factory, ambulance siren, and motorcycle, train). The overall accuracy of S1 decreased to 72% in the noisy environment from about 95% (i.e., within the noise-free environment). On the other hand, we confirmed that S2 improved the accuracy to 92% compared to S1 with the noise rejection method in a known noise-containing environment. We confirmed that it is difficult to provide high classification accuracy in an unknown noise environment.

### 4.3. Performance in a Noise with Ensemble-Based Approach

To verify the performance against various noise environments, classification accuracy was measured in three scenarios of S1 (baseline), S2 (denoising), and S3 (ensemble) for both known and unknown noise. In addition, U-Net training increased the known noise dataset to 3, 6, 9, 12, 15, and 30, and measured the accuracy. Figure 11 shows the performance results for each scenario according to the number of learned data types. Although the S1 (baseline) scenario, which was trained in a noise-free environment, was able to provide a classification accuracy of 95.33% in the noise-free environment, the performance decreased to under 71.65% in a noise environment. In scenario S2, to improve performance degradation in a noisy environment, the noise was reduced with U-net, and the classification model was applied. In this case, we confirmed that the more the known noise type was learned, the more the accuracy increased in the known noise environment. The classification accuracy was 92.27% for using up to 30 known noise types in the test. In the unknown noise environment, we confirmed that although the classification accuracy also increased as the number of learned data increased, it was lower (84.01%) than the accuracy in a known noise environment. Scenario S3 used an ensemble technique in addition to the U-Net-based noise rejection method of S2 to overcome the problem of accuracy degradation in an unknown noise environment. S3 also confirmed that the greater the number of data in the dataset of known noise, the higher the classification accuracy for known noise. Finally, we confirmed that 93.65% classification accuracy can be provided in the unknown noise environment, and thus the proposed S3 can provide high classification accuracy, not only in the known noise (95.33%) but also in the unknown noise (93.65%) environment, compared to S1 (70.74%) and S2 (84.01%).

Table 5 shows a comprehensive comparison of each scenario, S1, S2, and S3, trained by using 30 known noise types. We confirmed that the baseline scenario, S1, provided 71.65% of the total average classification accuracy in a noisy environment, and the S2 through U-Net with noise rejection method could provide 88.14% of the total average classification accuracy in both known and unknown noise environments. Finally, we confirmed that 94.59% classification accuracy can be provided by S3 compared to S1 (71.65%) and S2 (88.14%).

## 5. Conclusions

In this study, we proposed a method to classify the over-, normal-, and under-voltages supplied to the transformer using the acoustic signal of the transformer operating in various noise environments. The acoustic signal of the transformer was converted into an MS that mimicked the human auditory organ, and the classification model was designed based on the U-Net encoder model to effectively represent the important information of the acoustic signal. To ensure robustness to known noise, the noise was removed using a predefined noise dataset, and the deep learning model based on U-Net. Additionally, in unknown noise environments, the classification accuracy was improved by using an ensemble model. In the experimental environments, the testbeds were constructed using an oil-immersed power distribution transformer with a capacity of 150 kVA. As a result of the experiment, we confirmed that the proposed method can improve the under-, normal-, and over-voltage detection accuracy from 72 to 88 and to 94% (baseline to noise rejection and to noise rejection + ensemble), respectively, in various noisy environments.

## Figures and Tables

**Figure 1 sensors-22-01248-f001:**
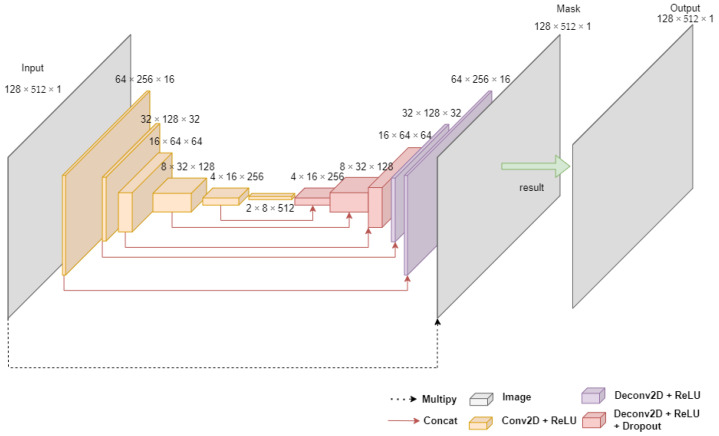
Overview of U-Net [25].

**Figure 2 sensors-22-01248-f002:**
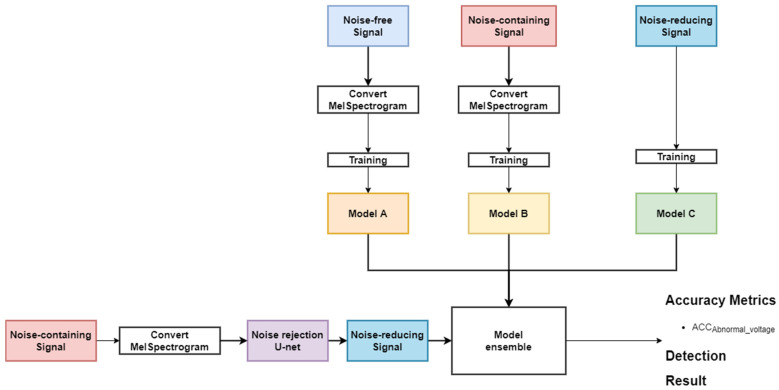
Overview of the proposed method EnsembleTransformerDet.

**Figure 3 sensors-22-01248-f003:**
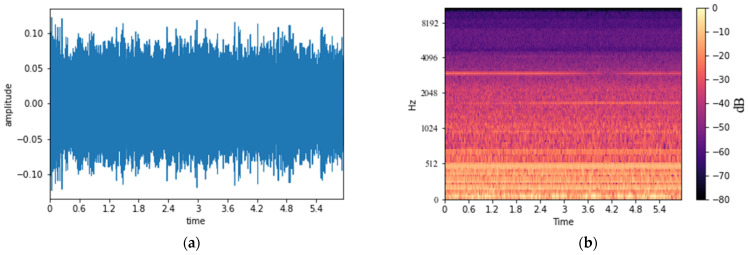
An example of the transformer acoustic signal with time- and time–frequency domains: (**a**) the acoustic signal of the transformer; (**b**) MS of the acoustic signal of the transformer.

**Figure 4 sensors-22-01248-f004:**
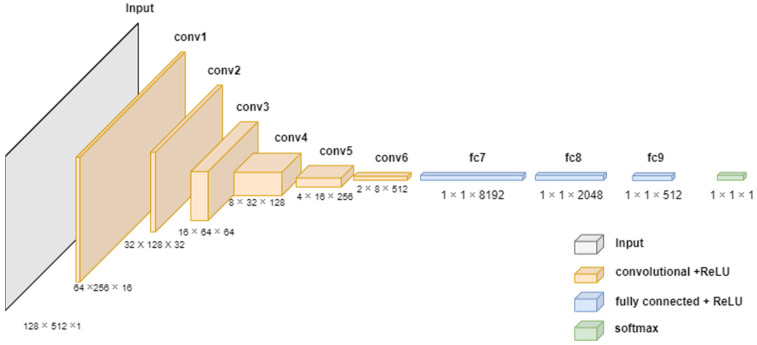
Classification model based on the U-Net encoder model.

**Figure 5 sensors-22-01248-f005:**
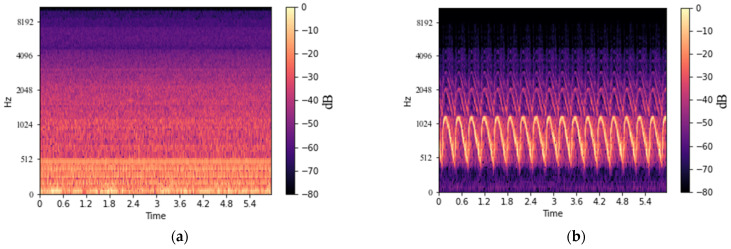
MS by noise rejection method: (**a**) transformer acoustic signal; (**b**) noise signal; (**c**) noise-containing signal; (**d**) noise-reducing signal.

**Figure 6 sensors-22-01248-f006:**
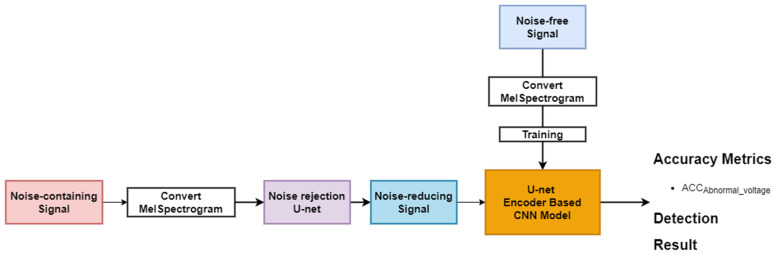
Noise rejection method with MS and U-Net.

**Figure 7 sensors-22-01248-f007:**
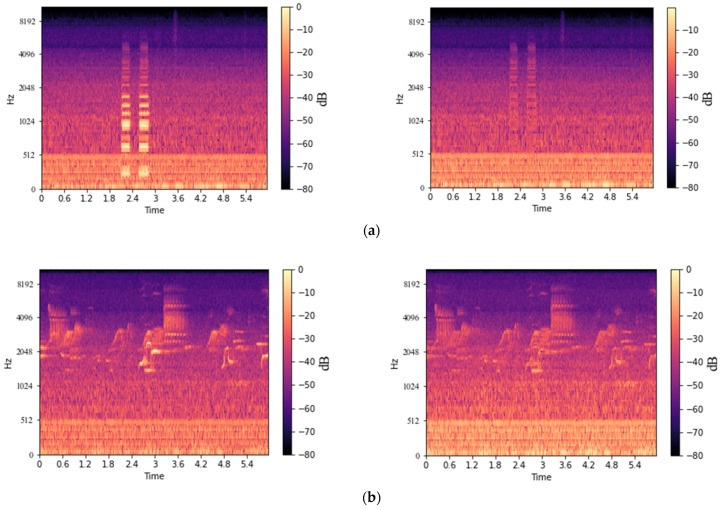
Examples of noise rejection result: (**a**) known noise; (**b**) unknown noise.

**Figure 8 sensors-22-01248-f008:**
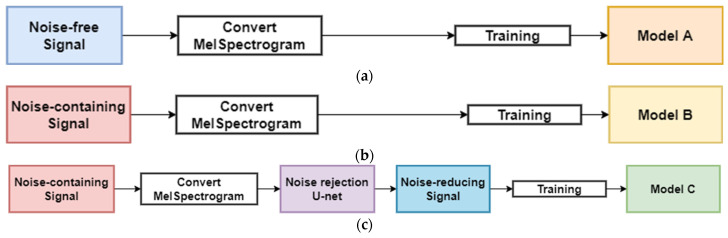
Each model for the ensemble method: (**a**) Model A with noise-free signal; (**b**) Model B with noise-containing signal; (**c**) Model C with noise-reducing signal.

**Figure 9 sensors-22-01248-f009:**
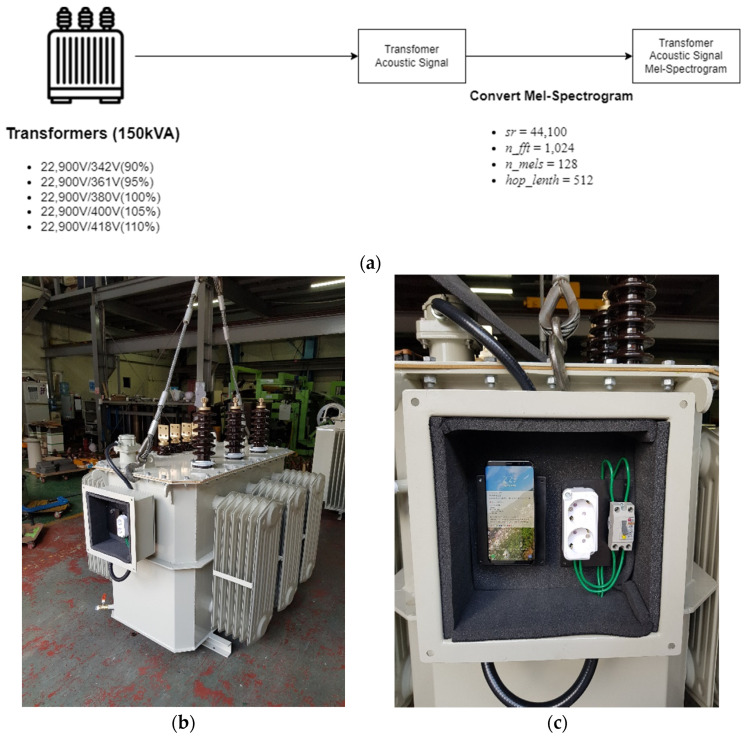
Experimental environments: (**a**) Overview of the experimental environments (**b**) Actual transformer to collect the acoustic signal (**c**) Attached smartphone on the transformer.

**Figure 10 sensors-22-01248-f010:**
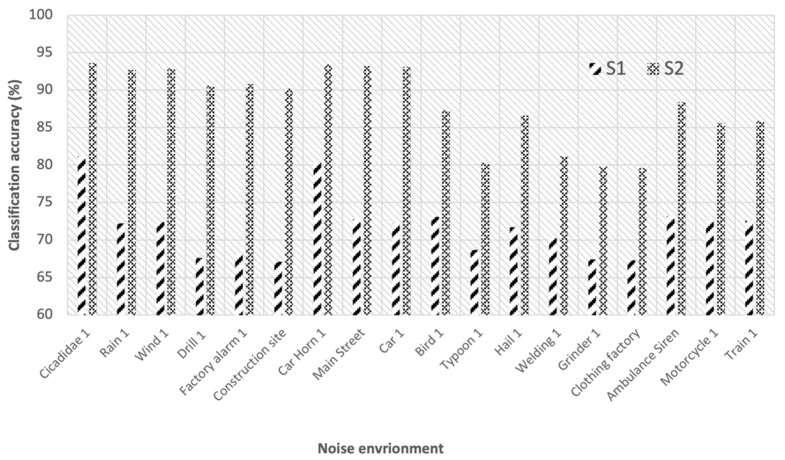
The classification accuracy in each noise environment.

**Figure 11 sensors-22-01248-f011:**
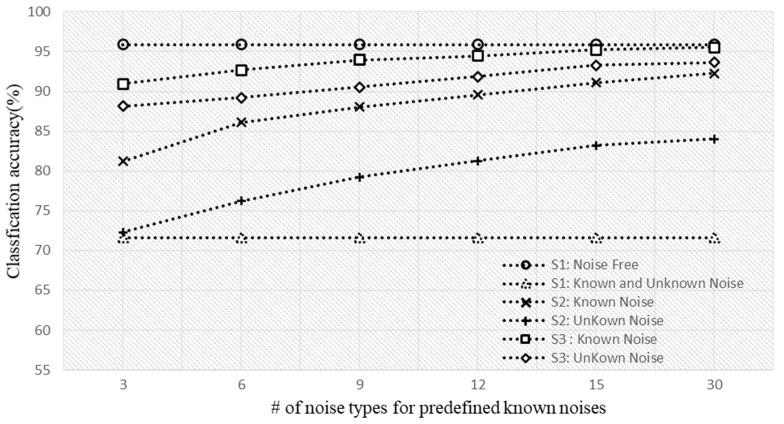
The accuracy in various noise environments according to the number of known noise types.

**Table 1 sensors-22-01248-t001:** The transformer core/winding diagnostics with vibration–acoustic methods [2,3,4,5,6,7,8,9,10,11,12,13,14,15,16,17,18,19,20].

Year	Non-Invasive Method	Considering Noise	Classification Method
[2,3,4,5,6,7,8,9,10,11,12,13,14] 2006–2018	X	X	Signal Processing or Visual inspection
[15,16,17] 2015–2018	O	X	Signal Processing
[18,19,20] 2019–2020	O	Known Noise	Machine Learning
Ours2021	O	Known Noise and Unknown Noise	Machine Learning

**Table 2 sensors-22-01248-t002:** Known and unknown noise datasets.

	Noise Environment	Noise Detail
Known	Nature	Rain 1, Rain 2, Rain 3, Cicadidae 1, Cicadidae 2, Thunder 1, Thunder 2,Wind 1, Wind 2, Wind 3
	Worksite	Excavator, Loader, Drill 1, Drill 2, Fork lift 1, Fork lift 2, Factory alarm 1, Factory alarm 2, Steel mill, Construction site
	City	Car Horn 1, Car Horn 2, Car Horn 3, Park 1, Park 2, Concert 1, Concert 2, Car 1, Car 2, Main Street
Unknown	Nature	Bird 1, Bird 2, Bird 3, Typoon 1, Typoon 2, Typoon 3, Hail 1, Hail 2, Frog, Cricket Chirping
	Worksite	Welding 1, Welding 2, Grinder 1, Grinder 2, Clothing factory, Building demolition, Air compressors, Breaker, Roller, Borewell drilling
	City	Motorcycle 1, Motorcycle 2, Ambulance Siren, Police Siren, Fire Truck Siren, Train 1, Train 2, Fire Work 1, Fire Work 2

**Table 3 sensors-22-01248-t003:** Each scenario for various noisy environments.

	Classification Methods	Learning Data	Noisy Environments
S1	Baseline	Noise-free	Noise-free
			Known and Unknown noise
S2	Noise rejection	Noise-free, Noise-containg	Known noise
			Unknown noise
S3	Ensemble	Noise-free, Noise-containg, Noise-reducing	Known noise
			Unknown noise

**Table 4 sensors-22-01248-t004:** Performance of noise rejection with U-Net (MSE).

	Noise Environment	Noise Rejection Performance with# of Predefined Known Noises		
		6	15	30
Known noise	Nature	0.30	0.61	0.72
	Worksite	0.39	0.62	0.72
	City	0.42	0.69	0.79
Unknown noise	Nature	0.22	0.32	0.46
	Worksite	0.22	0.39	0.41
	City	0.35	0.40	0.47

**Table 5 sensors-22-01248-t005:** Accuracy comparison of S1, S2, and S3 with 30 noise types.

Noisy Environment	S1	S2	S3
Known noise	72.56%	92.27%	95.33%
Unknown noise	70.74%	84.01%	93.65%
Total average	71.65%	88.14%	94.59%

## Data Availability

Not applicable.

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
