# Peer review of "Power Transformer Voltages Classification with Acoustic Signal in Various Noisy Environments"

_sensors, 2022, doi:10.3390/s22031248_

Round 1

Reviewer 1 Report

The manuscript “Power Transformer Voltages Classification with Acoustic Signal in Various Noisy Environments” by Mintae Kim and Sungju Lee presents a technique to classify the voltage levels in a transformer based on the analysis of emitted acoustic signals. The main advantage of the proposed approach is its comparatively high robustness to unknown sources of acoustic noise that has not been demonstrated by other similar solutions. This has been achieved by employing the noise rejection method with U-Net and extended datasets including the combinations of known and unknown sources of noise. The work is interesting and the presented results fit well in the scope of the journal. However, the referee cannot recommend it for publication due to the low quality of scientific writing and poor representation. Despite a concise and clear abstract that attracted the referee’s attention, the rest of the text is poorly written with multiple unnecessary repetitions of the text parts in various places that greatly complicate the reading process, hide the main message, and advantages of this study. The balance between the relevant and technical details is poor: the mentioned repeated parts take major space, while explanations of the modeling and experimental details are omitted resulting in non-reproducibility of the results that is a serious drawback for a scientific publication; many notations are not explained severely limiting the range of potential readers to specialists, whose number is expectedly very limited due to the applied character of this study. Therefore, the referee strongly suggests a major revision that should imply a complete re-writing of the text with eliminated repetitions, a clear statement of the problem, and analysis conditions (with an eventual figure/photo representing the analyzed system and positioning of the noise sources and measuring equipment), concise though the detailed formulation of the modeling results and conclusions. Finally, extensive editing of the English language and style is required.

Author Response

Thank you for your comments. We attach the modified manuscript and response letter according to your comments.

Reviewer 2 Report

From the article title point of view, the paper is well intentioned into  acoustic signal in various noisy environments. Congrats to the authors for aiming this research.

State of the art is relating to the current concern of the nowadays solution and the research interest area. From referances, there are more than 80 percent from recent researches.

  At chapter 4 the experimental setup is not satisfied presented. Does not offer sufficient basic information related to research proposed….  It cannot be used a a built in microphone of an cellphone in order to do detection of acoustic analysis … please take note about professional detection instrument patented for such a research, please visit this https://www.sqhead.com/industrial-measurements/condition-monitoring/

Mostly the built in microphone of an cellphone is not designed as an array microphone in order to improve the acousting signal caption, usually those type of microfones has their own noise compensation. So if your input accoustig signal is alter by the caption device, then every analysis done will be not suitable to some valid interpretation.

As a recommendation, for future research, try to repeat your experimental test by using a profesional device for accousting analysis, then compare it to you results from the present research and the you will understand the difference. 

At line 365, who is “Adam”… maybe is more suitable to use a reference…

The equation (6), where it come from???? Any reference aor at least a summary explaination of the parameters, otherwise must be exlude it.

Maybe if the article is reconsidered as to be presented without experimental comparation, it can be usefull for future researches. It does have some interesting ideas for considering the noise environment of a power transformer.

Maybe the paper can be published in the journal with some very major adjustments.

Author Response

(The authors gave the same response as above.)

Round 2

Reviewer 1 Report

The authors have addressed the raised scientific concerns, though the quality of writing must be improved further.

Reviewer 2 Report

The authors have replied to all the queries and concerns with proper references. However, to improve the connectivity for readers, authors can develop their research in future articles by use of professional acoustic device.

Good luck with future researches!